# Re-Envisioning an Early Years System of Care towards Equity in Canada: A Critical, Rapid Review

**DOI:** 10.3390/ijerph19159594

**Published:** 2022-08-04

**Authors:** Alison Jayne Gerlach, Alysha McFadden

**Affiliations:** 1School of Child and Youth Care, Faculty of Human and Social Development, University of Victoria, Victoria, BC V8W 2Y2, Canada; 2School of Nursing, University of British Columbia, Vancouver, BC V6T 2B5, Canada

**Keywords:** health inequity, structural inequities, structural violence, children, early child development, intersectionality, maternal wellness

## Abstract

**Background:** Many children in high-income countries, including Canada, experience unjust and preventable health inequities as a result of social and structural forces that are beyond their families’ immediate environment and control. In this context, early years programs, as a key population health initiative, have the potential to play a critical role in fostering family and child wellbeing. **Methods:** Informed by intersectionality, this rapid literature review captured a broad range of international, transdisciplinary literature in order to identify promising approaches for orienting early years systems of care towards equity in Canada. **Results:** Findings point to the need for a comprehensive, integrated and socially responsive early years system that has top-down political vision, leadership and accountability and bottom-up community-driven tailoring with an explicit focus on health promotion and maternal, family and community wellness using relational approaches. **Conclusion****s****:** Advancing child health equity in wealthy countries requires structural government-level changes that support cross-ministerial and intersectoral alliances. Employing intersectionality in this rapid review promotes contextualized and nuanced understandings of what is needed in order to advance a responsive, comprehensive and quality early years system of equity-oriented care. Further research is needed to prevent child health inequities that are disproportionally experienced by Indigenous and racialized children in wealthy countries such as Canada. olicy and research recommendations that have relevance for high-income countries in diverse global contexts are discussed.

## 1. Introduction

The character of a nation can be assessed by how it values its children and how this value is enacted in its social fabric and structure. The evidence informing governments is clear—the relationship between early adversity and lifelong well-being points to the critical role that early years systems play in supporting family and child well-being, including mitigating the impacts of structurally-rooted social determinants of health inequities [1]. The benefits of governments’ advancing and investing in early years systems have a compounding effect throughout children’s lifetime, their future children’s lifetime and society as a whole [2]. As Shonkoff [3] advocates, “all policies and programs that affect well-being during pregnancy and infancy present opportunities to promote lifelong health” (p. 2). Yet, in the face of irrefutable evidence, governments in many wealthy countries, including Canada, continue to fail to address the significant proportion of children whose start in life is shaped by the multifaceted downstream effects of adversity, including lack of food and safe housing [2]. Furthermore, a comprehensive and broader conceptualization of “the early years” as a key population health initiative has not received the political attention and action required [3].

This paper reports on the findings of a rapid literature review that sought to identify promising approaches to orienting early years systems of care towards equity in Canada. In this paper, “family” includes the unique cultural and kinship system of a particular child; family can include parents, caregivers, guardians, siblings, extended kin, kinship systems, social networks and community. The paper starts by positing intersectionality as an important analytical framework for examining health equity. The authors then hold up their home country of Canada as an exemplar of a high-income country in which child health inequities persist and remain inadequately examined and addressed, including in the early years sector. Following a description of the rapid review process and critical analysis of the findings, the authors provide a synthesis of emergent and inter-related themes on promising approaches to an equity orientation of the early years in the reviewed literature. The paper concludes with a discussion of policy and research recommendations that have relevance for high-income countries in diverse global contexts.

## 2. An Intersectional Framing of Children’s Early Years

How problems are framed shapes the direction for possible solutions. As will be elaborated in the methods section, the authors employed intersectionality as an analytical framework in this rapid review. Intersectionality, as coined by Black feminist scholars, promotes complex, historicized, contextualized and nuanced understandings of social injustices [4,5,6,7] and is conducive to examining health equity approaches [8]. Intersectionality is defined as:


*a way of understanding and analyzing the complexity in the world, in people, and in human experiences. The events and conditions of social and political life and the self can seldom be understood as shaped by one factor. They are generally shaped by many factors in diverse and mutually influencing ways. When it comes to social inequality, people’s lives and the organization of power in a given society are better understood as being shaped not by a single axis of social division, be it race or gender or class, but by many axes that work together and influence each other. Intersectionality as an analytical tool gives people better access to the complexity of the world and of themselves.*
[4]

Importantly, intersectionality disrupts neoliberal conceptualizations that health “disparities” are “an unfortunate and inevitable consequence of divergence from the behaviors and characteristics of the dominant population” [9], which can lead to parents and communities being blamed for and forced to manage structural disadvantages. Thus, an intersectional framing challenges the individualization of social problems and dominant positions within knowledge production that can keep systems of oppression and child health inequities intact [10]. For Indigenous families, newcomer families and racialized populations who have been living in Canada for generations—intersectionality can show how markers of identity related to ethnicity, language, culture and gender as they intersect with systemic racism, colonialism and White supremacy can prevent them from accessing social determinants of health and/or making effective use of public health and early years systems [11,12,13].

By employing intersectionality, the authors view children’s early health and well-being as complex, multidimensional and embedded in and shaped by multiple and inter-related social and structural factors and systems of power. In the Canadian context, intersectional analysis is inclusive of how broader social relations of power, including ongoing colonialism and systemic racism, shape communities and families’ everyday lives, their quality of life and their children’s chances of optimal health and well-being [14,15]. From this critical viewpoint, children are not inherently “vulnerable” to health inequities but rather their vulnerability is created by structural inequities and structural violence that result in particular child populations having a greater risk of health inequities. 

Currently in Canada, addressing issues of health equity in the early years has been inadequately examined [16]. The early years system is complex and fragmented, encompassing multiple sectors (e.g., health, education, child welfare and legal systems) and stakeholders (e.g., children, families, health care professionals, early childhood educators, social workers, legal experts and community leaders). Intersectionality provides an innovative and insightful approach to understanding how the complex interplay between multiple sectors and stakeholders is impacting health equity for young children and families and their equitable access to the early years system. This framing also seeks to make visible the complex relationships between broader socio-political forces and the subjective experiences of children and families in the context of their neighborhoods and communities [17]. Thus, an intersectional framing can better inform and support policy initiatives that acknowledge the multiple axes of inequity that define the everyday lived experiences of structurally marginalized and racialized communities and families [18].

## 3. Child Health Inequities in Canada

Health equity “means all people (individuals, groups and communities) have a fair chance to reach their full health potential and are not disadvantaged by social, economic and environmental conditions” [19]. Achieving health equity in the context of young children requires that *all* children (individuals, groups and communities) have a fair chance to reach their full health potential and are not disadvantaged by social, economic and environmental factors [20]. 

Across Canada, as in many wealthy countries, family well-being, maternal health and consequently children’s early health and development are shaped by a complex web of global, national and regional political and economic decisions and policies made by governments and public bodies [21]. The United Nations Convention on the Rights of the Child is a legal document that commits the Canadian government to fulfilling the rights of all children in Canada. Nonetheless, the historical and current political distribution of power and wealth in this country means that there are child populations that are particularly “vulnerable” to unfair, unnecessary and socially produced health inequities as a result of the social circumstances in which they are conceived, born, live and grow. As reported by UNICEF [22], Canada’s current policies and programs are not robust, equitable or sufficient to ensure every child’s rights are realized. 

Health inequities refer to differences in health between population groups that are socially produced, systematic in their unequal distribution across a population and are unnecessary, avoidable and unfair [23]. Child health inequities have persisted for generations in Canada—driven by upstream social, environmental and economic challenges, including systemic racism, substandard housing, limited access to health-promoting resources and socioeconomic deprivation [24]. Children at greater risk of health inequities in this country include children in lone-parent families led by women [25], children in the child protection system [26], Indigenous children [22,27], children with disabilities [28,29], children in families who experience racialization, stigma and/or social marginalization [30,31] and refugee children [32,33].

While Canadian children overall are doing quite well relative to children in low- and middle-income countries [22], UNICEF ranks Canada 30th among 38 high-income countries in terms of the well-being of children and youth under age 18. In spite of Canada’s wealth, child poverty in the country is pervasive [34]. The overall child poverty rate for children under six is 18.5%. However, this rate increases to 28.4% in the province of Manitoba and 34.4% (the highest in the country) in Nunavut [34].

UNICEF ranks Canada lowest in “child survival” [14]. A high prevalence of infant mortality stems from inequities in the social determinants of health and the impact of systemic discrimination [22]. Within the Canadian context, child mortality is a marker of extreme poverty and social exclusion and is disproportionally experienced by racialized population groups [22]. A health equity framing recognizes that child mortality, poverty and social exclusion are not genetic, biological or cultural but result from broader structurally-rooted determinants of health inequities that are often beyond families’ immediate control. The rate of infant mortality in regions with a higher concentration of Inuit families is 3.9 times higher than the general child population in Canada [35]. The high infant mortality rate in Inuit communities is an ongoing, downstream effect of economic exclusion, poverty, food insecurity and sub-standard housing as a result of Inuit peoples being forcibly displaced and dispossessed from their traditional lands by the Canadian government and forbidden to trap or fish for subsistence [36].

In 2008, the landmark commission on the Social Determinants of Health [37] outlined three overarching recommendations to address underlying structural causes of health inequities: (1) improving daily living conditions; (2) tacking the inequitable distribution of power, money and resources; and (3) measuring and understanding the problem and assessing the impact of action. However, as noted above, health inequities persist as a result of the maldistribution of power, money and resources [38,39]. Indeed, the United Nations Children’s Fund argues that it is not a country’s GDP but how those funds are distributed and allocated that impact child health and flourishing [22]. In other words, governments and society have created inequities for children, and it is within their power to address ongoing structural inequities. 

### Structural Inequities and Violence

Structural inequities refer to legislation, policies, prevailing discourses and power relationships in state institutions and systems that operate to create an inequitable distribution of determinants of health [40]. Structures like government policies and legislation coupled with status-quo social hierarchies create structural violence when they infringe upon individuals’ and population groups’ human rights, safety, agency and well-being—in other words, when they cause harm [41,42]. As violence is already built into the policy, discourse or legislative structure, structural violence may be challenging to trace and become imperceptible and taken for granted [43]. 

A key form of structural violence, and one that is often overlooked in early years programs, is structural racism [1,44]. “Structural racism assigns value and grants opportunities and privileges based on notions of ‘race’” (National Collaborating Centre for Determinants of Health, 2017. p. 23). Structural racism is embedded within a system of White supremacy that is “based on the presumed superiority of White racial identities” and “practices of Whiteness” which are assumed to be “the right way of organizing human life” (National Collaborating Centre for Determinants of Health, 2017. p. 24). Structural racism and White supremacy are pervasive in all aspects of Canada’s colonial society and its institutions, interacting with other forms of oppression to create inequitable social and health outcomes for racialized communities, families and their children [45]. 

The “toxic stress” that results from structural racism can reverberate across children’s life course [1] and accumulate across generations [46,47], negatively impacting their offspring’s academic chances, future employment opportunities and health into adulthood [48,49]. Dismantling structural racism is essential to eradicate child health inequities and implement children’s legislated rights [34]. In many wealthy, settler-colonial states, such as Canada, Australia and the United States (US), structural racism as it intersects with colonial violence can also influence Indigenous families’ engagement and participation in early years programs [50,51,52]. In Canada, the forced removal of Indigenous children from their families and communities through residential schools, “Sixties Scoop” and the provincial child protection systems [53,54], and the structural violence current child protection policies continue to cause have also understandably led many Indigenous families to be extremely protective of their children and reluctant to access any child-related services [52,55,56]. 

## 4. Methods

As a form of evidence synthesis, rapid reviews use a streamlined approach to accessing and synthesizing the literature on a focused topic or question within a shortened time frame [55,57]. Rapid reviews employ numerous combinations of strategies to conduct reviews and can include limiting the inclusion criteria, having one person screen the literature and not conducting quality appraisal in order to generate information within a short time frame [58]. The process used for this rapid review is outlined in the following section.

### 4.1. Step 1. Defining the Research Question

This review was part of a commissioned piece of work for a community stakeholder, the Office of the Representative for Children and Youth (RCY) in British Columbia (BC), to explore how the early years system in this province and Canada more broadly could be responsive to families with young children who experience a greater risk of health inequities. This review had a time frame of eight weeks and sought to address a question that was highly relevant to the RCY’s emergent strategic planning and decision making: *What are the recurring themes at systems, policy, program and practice levels in the early years sector that attend to child health equity issues?* This question was well aligned with the authors’ clinical practice and scholarship and their familiarity with the literature [13,16,59,60].

### 4.2. Step 2. Searching for Research Evidence 

Starting in April 2021, co-author McFadden scanned EBSCOHOST and CINAHL databases, Google scholar and the Canadian National Collaborating Centre for the Determinants of Health website (https://nccdh.ca/) (accessed on 1 April 2021) in order to identify health and allied research on child development, health equity and child health in English-language peer-reviewed and grey Canadian literature. Using Boolean methods, McFadden used key search terms including early years, childhood, early childhood, young children, early years programs, early intervention, equity, child health equity, child health inequity, child development and intersectionality. For example, one search included the terms childhood AND early intervention AND equity. Another search term included child health equity AND early years OR early childhood OR young children. 

This initial search highlighted a lack of Canadian published research, and in consultation with a librarian at the University of Victoria, the authors decided to expand the search to include academic and grey literature from a diverse, transdisciplinary body of English-language literature that had sufficient evidence of an equity-oriented focus or promising approach in other high-income countries, including the United Kingdom (UK), US, New Zealand, Australia and Sweden. Health equity is a relatively recent concept, and it was advised by the university librarian not to use date parameters in the search. Furthermore, intersectionality underscores that diversity of knowledges is essential to fully understand the inherent complexities of inequities and injustices (Collins, 2019). In taking up this viewpoint, all forms of knowledge, including reports, randomized control trials and peer-reviewed academic articles, were included in the search. 

Using an iterative approach, the authors reached consensus on the inclusion criteria as summarized in Table 1. McFadden screened titles and abstracts of all articles identified in the broader search that included international literature. This process was guided by the inclusion and exclusion criteria. McFadden then imported the search results into RefWorks, a referencing management system, to facilitate screening titles, abstracts and duplicate removal. McFadden completed the literature search in June 2021.

As summarized in Figure 1, the preliminary search generated 1304 citations, with 671 citations meeting the criteria for review. McFadden undertook full-text reviews on 671 publications resulting in 130 publications that met all inclusion criteria. 

### 4.3. Step 3. Data Extraction

Data extraction was guided by the research question, and the findings were synthesized and organized according to the following four questions that were agreed upon by the community stakeholder and authors: (1) How is intersectionality being used to help understand health equity in the context of early years programs? (2) How is child health equity being conceptualized/framed in current literature and in different wealthy global contexts? (3) How are early years programs, policies and practices responding to health equity/inequity? (4) Are there any gaps in knowledge? 

Data extraction was performed by one reviewer McFadden and checked by a second reviewer Gerlach, and consensus was reached through discussion. It is beyond the scope of this paper to summarize all 130 manuscripts that were reviewed; a sample of papers from diverse global contexts is included in Table 2. This table highlights that despite the heterogeneous and transdisciplinary nature of the publications reviewed, each manuscript had clear elements of an equity orientation. McFadden organized the findings into an annotated bibliography which included what was known and what was absent in the literature in relation to each of the four guiding questions. At this stage, both authors and the community stakeholder engaged in multiple discussions to determine what evidence was helpful and relevant to addressing the primary research question. 

### 4.4. Step 4. Analyzing the Findings

As noted above, the inclusion of quality appraisal in rapid reviews is variable and was not undertaken for this review [58]. Rather, the authors, with the support of the community stakeholder, used intersectionality as a critical framework in appraising and analyzing the findings in order to identify emergent, recurring themes in the literature. In taking this critical stance, it is important to note that the authors’ analyses were invariably shaped by their personal and professional experiences and positionalities. The authors both identify as White settler, cis-gender, privileged women and mothers currently living in western Canada and with extensive practice backgrounds in pediatric occupational therapy (Gerlach) and public health nursing (McFadden) with diverse families and children that experience a greater risk of health inequities. Gerlach has also been involved in researching equity issues in the early years sector for the past 20 years [16,61,62,63,64].

## 5. Findings

As noted above, the initial search resulted in a total of 1304 publications. After removing publications that did not meet the criteria, McFadden undertook full-text reviews on 671 publications resulting in 130 publications that met all inclusion criteria. In the following section, the authors describe recurring and inter-related themes identified in the reviewed literature and which they contend hold promise for transforming and orienting early years systems towards equity in Canada and have relevance for similar wealthy global contexts.

### 5.1. Top-Down Political Leadership—Upstream Accountability and Macro-System Actions

If child health inequities in early childhood are viewed as being primarily a result of children’s exposure to and experiences of varying qualities of material living conditions that are shaped by economic and political structures and their justifying ideologies—a key health promotion approach is that child health inequities can be reduced by moving upstream and influencing structurally rooted causes [65]. “Upstream interventions and strategies focus on improving fundamental social and economic structures (that is the root causes of health inequities) in order to decrease barriers and improve supports that allow people to achieve their full health potential” ([7], p. 6). Increasing access to existing early years services will, therefore, not have a significant impact on the upstream socio-economic and political structures that create child health inequities [66]. Rather, a foundation for an equity orientation is political leadership, vision and accountability at all levels of governance [37,38]. In other words, government and intersectoral actions and policies are needed that challenge the status quo [67,68,69] and shift existing economic and political structural inequities, particularly as they impact historically marginalized communities in Canada.

Canada, like the US, Australia and New Zealand, is a colonial state with a history of genocide against generations of Indigenous communities, families and children [54]. The historical and ongoing, multifaceted forces and impacts of colonialism mean that Indigenous children in many colonial states experience poorer life opportunities and health inequities compared to the broader child population [27]. In Canada, government early years policies are primarily centred on individualistic and developmental perspectives of childhood, with childcare and optimal child development viewed as a commodity and a means to economic productivity and prosperity [70,71]. This hegemony disregards Indigenous knowledges and Indigenous determinants of child health that include cultural identity, ancestral languages and connections to the land [47,72,73] and conflicts with Indigenous views on raising young children, including children with neurodiversity and developmental disabilities [74]. Indigenous perspectives often underline that community members, including every child, are intrinsically valued and have something to contribute to the well-being of the community [74].

In settler-colonial states such as Canada, achieving health equity for Indigenous children requires a transformative shift to uphold Indigenous rights to self-governance and determination of programs with Indigenous families and children [75]. In this context, top-down leadership means Indigenous self-governance to ensure leadership, control and direction over the best interests of Indigenous families and children [76]. In Canada, the right to self-governance is legislated in the 2007 United Nations Declaration on the Rights of Indigenous Peoples [77] and in Bill C-92. As of 1 January 2020, Bill C-92 has received Royal Assent and is now law in Canada. This law affirms that Indigenous peoples across Canada have jurisdiction over child and family services. Bill C-92 has principles that are applicable within all national, provincial and territorial levels, including the best interests of the child, cultural continuity and substantive equality [54]) in 2020 [78]. Indigenous people’s right to self-governance is also called for in the 2015 Truth and Reconciliation Commission of Canada [54] and the 2018 National Indigenous Early Learning and Child Care Framework [76].

In the context of wealthy global countries, the need for top-down political vision and leadership that engages all sectors of government is also evident in international calls for “health in all policies” (HiAP) as a systematic approach to recognizing and reducing social and health-related harms from contemplated policies [79]. A HiAP approach recognizes that tackling the social and structural determinants of health requires action from outside of and alongside the healthcare system [80]. HiAP aims to ensure that all government sectors assess how any proposed policy will directly or indirectly affect the upstream determinants of health that may impact health equity [81]. For example, the health impacts of zoning regulations that may increase urban sprawl will be assessed for their effect on increasing fossil fuel consumption, air quality, pollution levels and global climate change [81]. 

### 5.2. A Comprehensive and Responsive System for Community, Family and Maternal Wellness

Pre-conception, prenatal and early years mark a window of both opportunity and vulnerability for, often intergenerational, child health inequities [3,67,82]. From a population health perspective, ‘health’ is viewed as the physical, spiritual, mental, emotional, environmental, social, cultural and economic wellness of the individual, family and community [7,83]. Given that early child development is a key determinant of population health, a comprehensive early years system of care, when structured from prenatal through to age eight, has the potential to be a foundational populational health initiative and a critical entry point for family and community wellness and consequently children’s health equity [67]. As discussed in the following section, advancing child health equity requires a political reframing to advance an integrated and comprehensive system that responds to community, family and maternal wellness [3]. 

Early years systems in Canada often comprise a complex and fragmented network of child-centric programs and services aimed solely at the level of individual families and children. Canadian and international, Indigenous and allied perspectives highlight that the maternal/child discourse, and concomitant siloing of services for women and children, necessitate a needed shift to address the mutuality between maternal, family and community wellness [16,46,47,73]. Thus, an essential starting point in avoiding or reducing child health inequities is a focus on health promotion in an integrated early years system that includes tailored approaches to fostering community, family and maternal wellness. Sweden is an exemplar of a country that has a national program that targets prenatal, maternal, newborn and early child health through a continuum of social policies and care into school-based health promotion and has some of the best health indicators for maternal and child health and well-being in the world [82].

This comprehensive approach to health promotion contrasts with dominant, neoliberal discourses that tend to focus on individual lifestyles or health behaviors [65] which can lack relevance and efficacy for families raising children in structurally marginalized neighborhoods [84]. Moreover, from an equity perspective, health promotion is not merely about educating people to change their behavior but of equal importance is changing the conditions under which families can lead healthy and fulfilling lives [85]. A comprehensive early years system would thus provide holistic, intersectoral and wrap-around support for communities and families who are experiencing health-promoting challenges such as food or housing insecurity, parental mental health problems, violence, structural racism and discrimination [14,16,86,87]. Health promotion strategies and initiatives would also be targeted and tailored in response to unique family contexts, including the social, economic, cultural and historical determinants that may limit or prevent access for newcomer and racialized families from effectively accessing early years services, resources and supports [18]. 

This broader and inclusive conceptualization of health promotion is evident in how Australia expanded upon the “First 1000 days” movement https://thousanddays.org (accessed on 1 May 2021) which highlights the importance of pre-conception and infant nutrition on child health and development during the first 2 years of its life. This nutrition and maternal care program acknowledged that “improved nutrition alone will not address the current poor health and well-being status of Indigenous children in Australia and elsewhere” [88]. The program was thus reconceptualized to address Indigenous community and family wellness and increase antenatal and early years’ engagement, access and service use. Through a community governance strategy and visioning, the program also sought to reduce health inequities by expanding the model to include community economic development through micro-business solutions with Indigenous families to generate new sources of income and facilitate community self-determination [88].

Prioritizing health promotion and fostering community and family wellness, while typically not strong features in early years programs [44,68], are often evident in Indigenous early years programs, reflecting a recognition of the inextricable continuities between community, family and child wellness [89,90]. Across Canada, Friendship Centres are an example of an integrated whole family wellness and community hub model, offering a broad range of co-located social services and programs with urban Indigenous families and children. An increasing number of Friendship Centres operate early child development and child care programs [91], acting as a “hook” for mobilizing community involvement for young families and as a hub for meeting a range of services and social support needs of community members [92]. Social pediatric models are also emerging in a small number of socio-economically marginalized neighborhoods in Canada and advance child health equity by focusing on intersectoral activities that support community and family wellness [93,94]. 

### 5.3. Coordinated and Funded Intersectoral Alliances and Actions

Prioritizing health promotion and wellness, as outlined in the previous section, and responding to the social complexities of families’ lives, requires actions on the social determinants of maternal, child, community and family health and wellness by funded and coordinated intersectoral alliances [2,95]. Intersectoral alliances can include adult and child health, early years, transportation, housing, food insecurity sectors and so forth that collaborate on advancing social determinants of health (Public Health Agency of Canada, 2021). For example, in the US, intersectoral medical and legal advocacy and actions reduced asthma-related pediatric hospitalizations by requiring landlords to reduce household mold [96]. Internationally, there is also promising evidence of alliances between the early years and education sectors which are often funded and administered by different branches of government. Evidence from the US and UK point to how funded intersectoral alliances and actions between the early years and the “early grades” of the education sectors can advance the health, academic and life chances of children growing up in socio-economically disadvantaged neighborhoods and families [86,97,98,99]. These approaches emphasize the need for a continuous and cohesive system of care—supported by funding and policies focused on family strengths and shared priorities [99,100]. 

### 5.4. Embedding Equity in Data Collection and Accountability Systems

Advancing child health equity also requires increased investment in relevant outcome measures and sustainable data collection and monitoring activities at all levels of government in order to provide a robust understanding of the prevalence of child health inequities and how upstream structural inequities, including poverty and systemic racism, differentially impact child populations and family quality of life. Family quality of life has been defined as “a dynamic sense of well-being of the family, collectively and subjectively defined and informed by its members, in which individual and family-level needs interact” ([89], p. 262) [2,97,101,102,103,104]. From an equity-oriented perspective, the focus is on generating data about progress on the upstream, structural and social determinants of health inequities [38], including measuring the impacts and wellness outcomes of intersectoral actions [105].

However, the nuanced, contextualized and intersectoral nature of an equity-oriented early years system of care can make it challenging to measure its impact [106]. Early years programs are typically focused on and/or funded based on service or program outputs rather than family or child health outcomes; therefore, they tend to persist with service delivery methods that may not be optimally effective [107]. Consistent with an equity lens, data collection needs to be community-informed, context-specific and strengths-based with multiple measures of health and well-being [88], including children’s views [108]. There is also a need to embed equity in government monitoring and accountability systems, including quality improvement [91,97]. Importantly given the historical and ongoing colonial violence perpetrated by the Canadian state, political transformation to uphold Indigenous governance over data collection, ownership and its application is crucial [109].

### 5.5. Bottom-Up Demand—Community Driven Co-Design and Tailoring

In order to address the upstream causes of child health inequities, the top-down vision and accountability outlined in the previous sections need to be equally matched with actively engaged local communities and families in the design of inter-/cross-sectoral collaborations and enhanced service integration so that service systems are driven by and responsive to the lived realities and priorities of local communities and families [3,96,101,110]. Community-driven and participatory health promotion approaches are strengths-based, mobilize and build on community and lived knowledges, expertise and resources and focus on building and maintaining enduring supportive relationships and community connectedness [88,94,95]. Central to an equity orientation, participatory approaches ensure that any actions are tailored and adapted so that they fit and work for local community and family contexts, meeting self-identified emergent needs and priorities and allowing for experimentation [16,106,111,112]. Scaling-up standardized programs that are not contextualized to the local community rarely work and can widen child health equity gaps [66,96]. Community participation in early years policy development, which includes multi-sectors and levels of governments (national, regional and municipal) and local partners and stakeholders (i.e., children, youth, caregivers and local leaders), can foster a strong sense of ownership in the process and outcomes, informing and increasing the implementation of impactful and sustainable early years policies and new initiatives [95,108,113].

In concert with community-driven and tailored approaches, there is growing interest in “place-based” programming to address child health inequities in wealthy countries [66,95,114]. Features of place-based approaches include identifying and building on existing community strengths and resources, improving service delivery and coordination, increasing social networks and working towards particular social objectives aimed at improving and empowering whole neighborhoods [87]. Successful, community-based and driven place-based interventions involve having a clear and collaborative governance structure that allows different levels of government, different government departments, non-government organizations and communities to come together to develop and implement comprehensive place-based action plans [114]. Indeed, many of the elements found in the health equity literature, as outlined in this paper, are characteristic of placed-based approaches, including strong community participation and governance, intersectoral partnerships and collaborative agendas, a strengths-based and relational orientation to working with communities and families, community tailoring and efforts focused on addressing the upstream causes of child health inequities with whole neighborhoods and communities [95,115].

### 5.6. Relational Approaches

In moving beyond individualistic and decontextualized notions of children’s early health and development, and aligned with the community-driven approaches outlined above, a relational (re)framing of early years programs means that they have the capacity to understand and respond to the social complexities that underlie child health inequities within diverse family and community contexts [3]. Thus, in contrast to a standardized or a one-size-fits-all approach, and aligned with the aforementioned elements of an equity-oriented system of care, relational approaches require that service agencies and providers prioritize time to learn about, and respond and are accountable to, the lived realities and priorities of the families and children they are serving [16,95,106].

A relational framing of early years programming is not a quick fix or intervention program. Similar to and aligned with place-based approaches, the capacity to implement a relational approach necessitates that government and community organizations and programs invest time and resources in building and maintaining enduring and supportive relationships with communities, families and children in order to learn with and from them about how to adapt and provide meaningful early years programs that build on their strengths and respond to their priorities and lived realities [94]. However, its implementation can be challenging when service providers lack the resources to spend time fostering relationships as an essential starting point for meaningful and effective services [116]. Thus, political and organizational leadership buy-in and support are fundamental for this approach to be actualized [86].

In the equity-aligned international literature, there are also specific philosophical approaches to understanding, designing and delivering programs with families and children that are distinct in their origins and focus but share a relational orientation. These include anti-racist programming [44,104,117], cultural safety [11,118] and trauma-informed [94,101,119], or trauma- and violence-informed approaches [61]. This relational orientation to early years programs and practices is predicated on funders’, managers’ and service providers’ understanding of and capacity to respond to how structural inequities and structural violence are differentially impacting families and young children [67,116].

## 6. Discussion of Policy and Research Recommendations

Fundamental to revisioning an equity-oriented early years system of care in Canada is revealing and addressing how multifaceted social factors and structural inequities can cause child health inequities. Employing intersectionality in this rapid review provided a nuanced, critical analytical framework to help the authors unpack the complexities of community, family, maternal and child health and well-being and what needs to be addressed in order to advance a comprehensive and quality early years system of care with the capacity to prevent “vulnerable” young children from experiencing preventable health inequities with potentially lifelong consequences.

As the findings show, policy changes are needed to support relational approaches, so that community organizations and early years providers have the mandate and capacity to adapt their programming so that it is tailored for/with and responsive to diverse community and family contexts [3,16,95,106]. Moreover, whilst the relational approaches outlined in the findings are often framed in relation to communities and families who experience structurally-rooted forms of social disadvantage, including newcomer and Indigenous families [18,118], the authors question why these approaches are not considered ethical practices in all settings, with all families and children.

Central to advancing child health equity in an early years system of care is that “care” encompasses maternal health, the early years and the early grades in school. As the findings highlight, broadening the scope of an early years system of care requires a transformative political shift and a radical re-envisioning of programs and services shifting from a focus on “the child” towards a wider range of integrated, timely and accessible whole family services that connect prenatal care with a broader reframing of early years programs focused on well-being and extending into the early grades at school [3,97]. Evidence from diverse high-income countries shows that community-informed and -driven care in an integrated and comprehensive early years system needs to include holistic, wrap-around supports and health promotion approaches that are tailored by the community for the community and have the capacity to foster maternal, family and community wellness [3,16,46,47,73,82]. International literature also emphasizes that improving access to determinants of health requires political ownership and leadership in concert with a cohesive vision of child health goals, government-level coordination across ministries and systematic intersectoral or cross-sectoral coordination, collaboration and problem-solving [2,44,95,120,121,122].

In Canada, there is also a need to have a meaningful HiAP action plan at federal and provincial/territorial levels of government [81]. In the context of advancing health equity with Indigenous communities, families and children, political action on Indigenous rights to self-governance and the recommendations that have been put forward in good faith by Indigenous communities and leaders need to be upheld and implemented [54,77,78]. This includes enacting the recommendations arising from a national Indigenous Early Learning and Child Care Framework [76], the underlying principles of which are well aligned with the findings of this rapid review.

Increased attention to child populations in Canada that are most vulnerable to health inequities as a result of the COVID-19 pandemic [123] may provide a catalyst for top-down political leadership and actions on international, national and provincial recommendations that are waiting to be implemented to ensure that all children in this wealthy country can have the best start in life [22,31,34,124]. Since this review was undertaken, there has been promising political action by the provincial BC government in terms of embedding equity in data collection and accountability systems with the introduction of a new anti-racism data act [125] which was co-developed with Indigenous leaders. However, once legislated, it remains to be seen if/how this act will become integrated into the provincially-funded early years system.

Overall, the findings show a paucity of research on addressing entrenched and long-standing child health inequities in wealthy, global jurisdictions, including an equity framing of services for children with disabilities and/or neurodiversity who are also “vulnerable” to health inequities. Research on the experiences and voices of parents, children and youth on a comprehensive, inclusive and responsive early years system of care is needed. Moreover, there is a need for research on the impacts and wellness outcomes of intersectoral actions and distinct place-based and land-based approaches. The authors contend that the elements of an equity-oriented early years system, as highlighted in this paper, are also highly relevant to undertaking much-needed equity-oriented research.

## 7. Limitations

There are limitations to this scoping review. As discussed, the authors did not appraise the quality of the studies included in this review. The authors have also been transparent about the inseparability between their own social locations and positionalities and their critical analysis of the findings.

## 8. Conclusions

A comprehensive and broader conceptualization of “the early years” as a key population health initiative has not received the political attention and action required. The findings of this rapid review show that orienting an early years system of care towards equity in Canada, and similar high-income countries, requires political leadership at all levels of government in concert with community knowledge and participation. Such a system must be inclusive of and function at the intersections of community, family and maternal wellness and children’s early health and life chances. These efforts cannot be carried by any one government branch or organization. In settler-colonial states such as Canada, advancing health equity with Indigenous communities, families and children starts with the legislated rights of Indigenous peoples to self-governance and determination to be put into transformative action.

The challenge for governments in advancing child health equity is that it cannot be achieved without structural changes that support community-driven and place-based approaches that are informed by and tailored for local community strengths and priorities, cross-ministerial and intersectoral alliances and actions that allow for community-driven and tailored, relational approaches to programming. Greater attention and research are needed to act on unjust and potentially preventable child health inequities that continue to be disproportionally experienced by Indigenous and racialized children in wealthy countries such as Canada.

## Figures and Tables

**Figure 1 ijerph-19-09594-f001:**
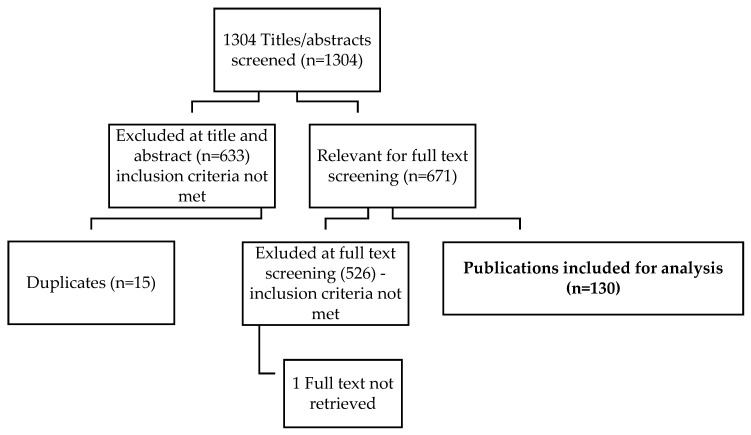
Summary of Search Process.

**Table 1 ijerph-19-09594-t001:** Inclusion and Exclusion Criteria.

	Inclusion Criteria	Exclusion Criteria
**Population**	Prenatal period Children aged 0–8 years (In the authors’ home province of BC, the early years is currently conceptualized from birth to 8 years of age)	The literature that exclusively focused on individuals over the age of 8 (e.g., youth, non-pregnant women)
**Global Setting**	High-income countries	Low- or middle-income countries
**Research**	Explicit equity focus Critical and intersectional framing of health in/equity Prenatal to age 8 years Written in English language Any study type	Lack of equity focus No evidence of critical or intersectional framing of health in/equity Children over 8 years of age Written in language other than English

**Table 2 ijerph-19-09594-t002:** Sample of Final Articles that met the Inclusion Criteria in Alphabetical Order.

Author(s) (Year) and Title	Location	Publication Type and Research Method(s)	Target Population	Findings	Elements of Equity Orientation
Archambault, J., et al., (2020). *Early childhood education and care access for children from disadvantaged backgrounds: Using a framework to guide intervention*	Canada	Peer-reviewed publication reporting on literature synthesis	Children 0–5 years and families from “disadvantaged backgrounds”	Authors propose a framework identifying factors influencing access to quality early childhood education and care for children from “disadvantaged backgrounds”.	Multi-purpose and co-located; intersectoral and multisectoral partners and actions; integrated service; relational and responsive programming; government buy-in and support, neighborhood-level programs; outreach; family as partners; training support for staff.
Ball, J. (2005). *Early childhood care and development programs as hook and hub for inter-sectoral service delivery in First Nations communities*	Canada	Peer-reviewed publication reporting on multi-site, mixed methods study	First Nations families and young children	Author proposes a conceptual model of early childhood care and development programs as a hook for mobilizing community involvement in supporting young children and families and as a hub for meeting a range of service and social support needs of community members.	Co-location of child care with other services in multi-purpose, community-based service centres to improve access to health monitoring and care, screening for special services and early interventions.
Baum, F., et al., (2020). *Creating political will for action on health equity: practical lessons for public health policy actors.*	Australia	Peer-reviewed publication reporting on qualitative case study methodology	Policy stakeholders	Paper provides evidence of the factors that work for or against action to reduce health inequities by addressing the social determinants of health inequities.	Political framing of inequities away from a medical and behavioral framing and towards human right to health.
Beck, A. F., et al., (2019). *Cooling the hot spots where child hospitalization rates are high: A neighborhood approach to population health*	United States	Peer-reviewed publication on quality improvement initiative	Hospitalized children from lower-income families	Hospitalizations reduced by 20% through intersectoral action, multi-disciplinary teams, and community participation; and use of actionable, real-time data.	Intersectional action; community participation and tailoring; neighborhood level; data-driven action.
Berry, O. O., et al., (2021). *Social determinants of health: The impact of racism on early childhood mental health*	United States	Peer-reviewed publication synthesizing published literature and longitudinal studies	Racialized children ages 0–5 and their caregivers	Young children’s socio-emotional development is highly influenced by exposure to multiple and interconnecting levels of racism and discrimination.	Relational and anti-racist prevention and intervention strategies targeting young children and parents.
Boulton, A. F., et al., (2014). *Whānau ora; he whakaaro Ā whānau: Māori family views of family* *wellbeing*	Aotearoa New Zealand	Peer-reviewed publication reporting on qualitative study and policy analysis	Māori families	Whānau ora (family well-being) is a multidimensional concept that is time and context specific. Requires Māori self-determination, long-term relationships and financial investments.	Holistic, wrap-around, intersectoral services for whole family. A “one-size fits all” approach is ineffective. Flexibility needed for service providers to work across sectors to manage complex social problems.
Boone Blanchard, S., et al., (2021). *Confronting racism and bias within early intervention: The responsibility of systems and individuals to influence change and advance equity*	United States	Discussion paper in peer-reviewed publication	Children 0–5 years	Policies needs go beyond maternal-infant health policies and include the early years of life. Need a health in all policies framework that includes employment, family leave, social systems and health care. Focus on fixing the system and not the child.	Participation and partnerships; anti-racism and anti-oppression practices and policies; trauma-informed approaches; anti-racism and anti-bias training; accountability systems; governance and leadership in social and public policies.
Dodge, K. A. (2018). *Toward population impact from early childhood psychological interventions*	United States	Peer-reviewed publication synthesizing published literature and empirical research	Children 0–5 years	Services need to align with children’s needs and evidence-based services need to be readily available with improved continuity between services. Need to catalogue community programs to find where gaps exist.	Political buy-in and ownership; accountability systems through data collection and action; Place-based approach; combine top-down approach to improve determinants of health and neighborhood or local level targeted community resources; tailor to local contexts; data tracking and accountability systems.
Gerlach, A. J., et al., (2018). *Relational approaches to fostering health equity for Indigenous children through early childhood intervention*	Canada	Peer-reviewed publication reporting on qualitative study	Indigenous parents and early child development providers in urban centres	Relational perspective of family well-being and relational approaches to early child development programming	Inseparability between family well-being and child health equity; socially-responsive and tailored relational approaches and broader scope of practice.
Early Intervention Foundation (2020). *Adverse childhood experiences: What we know, what we don’t know, and what should happen next*	United Kingdom	Research report	Children, young people and families	Ongoing misconceptions about adverse childhood experiences. There are no quick fixes and need for comprehensive public health approaches in local communities.	Comprehensive system to support healthy communities and families; early years needs to extend into educational system.
Janus, M., et al., (2021). *Population-level data on child development at school entry reflecting social determinants of health: A narrative review of studies using the early developmental instrument*	International	Peer-reviewed publication reporting on narrative review	Children 0–5 years	The Early Development Instrument (EDI) is an effective tool for monitoring children’s developmental health and increasing understanding on impacts of adverse social determinants. Universal interventions may not be effective at meeting the needs of children with increased neighborhood-level adversity and/or in socio-economically marginalized families.	Holistic and neighborhood-level, intersectoral interventions to address social determinants of health.
Hickey, S., et al., (2021). *A call for action that cannot go to voicemail: Research activism to urgently improve indigenous perinatal health and wellbeing*	Australia, Aotearoa New Zealand, United States, Canada	Discussion paper in peer-reviewed publication	Indigenous families	Urgent need for adequately funded Indigenous-led solutions to address perinatal health inequities for Indigenous families in high-income settler-colonial countries.	Privileging of Indigenous knowledges and solutions; Indigenous governance; continuity of care; focus on family well-being; strengths-based; improving “cultural capabilities of non-Indigenous staff”.
Loock, C., et al., (2020). *Social pediatrics: A model to confront family poverty, adversity, and housing instability and foster healthy child and adolescent development and resilience*	Canada	Book chapter on social pediatrics model	Children and families from structurally vulnerable, low-income communities	Social pediatrics model involves primary care clinic, specialty outreach and legal aid through place- and strengths-based, localized care with emphasis on horizontal partnerships and communication. Effective at providing holistic care from prenatal to child to youth to families within the context of the community.	Integrated management and team approach; responsive and relational care; neighborhood-level access; intersectoral support; shared decision making; bottom-up demand; place-based approaches; validation of community-based knowledge and expertise; child-led, community-driven responses.
McBride, D., et al., (2021). *Family hubs, Stockton-on-Tees: Early childhood services case example*	United Kingdom	Report on case study example	Children 0–19 years and families	Family hubs require strong leadership and visioning to provide whole-family support that builds on existing relationships and communication and embody core values such as respect, inclusiveness, honesty, compassion, cooperation and humility.	Wholistic family hub model (integrated care, community-level needs, tailored programs); case management; universal programming and targeted programming; outreach services; systems navigation; relational and reflective practices; strong leadership and vision; strengths-based; build upon existing relationships and intersectoral partners.
Richter, L. M., et al., (2017). *Investing in the foundation of sustainable development: pathways to scale up for early childhood development*	International	Discussion paper in peer-reviewed publication	Young children and families	Data is needed to monitor the implementation of policies and requires multiple forms of knowledge and expertise, intersectoral partnerships with government and policy makers and mobilization of parents, families and communities. Calls for United Nations Special Advisor for Early Childhood Development as a way to put the issue high on political agendas, facilitate coordination and promote accountability.	Intersectoral action; community-led and driven; political buy-in and top-down leadership and governance; holistic continuum of care from prenatal to adolescent and women’s health; outreach; political buy-in and need better research and data-driven evaluations; flexibly adapted at the local level with sharing of responsibility; health in all policies; monitor adoption of and implementation of policies and funding; build local capacity.
Ritte, R., et al., (2016). *An Australian model of the First 1000 days: An Indigenous-led process to turn an international initiative into an early-life strategy benefiting Indigenous families*	Australia	Discussion paper in peer-reviewed publication	Preconception to early years	Empirical evidence needed for the future well-being of future generations. Need for community-informed, strengths-based data and decolonizing research and methodologies, community governance; cultural responsiveness and cultural safety. Need to build capacity of families and healthcare and allied workforce.	Intersectoral action; community participation and co-creation and leading; integrated services; health promotion holistic focus on health and wellness including a focus on families and communities; strengths-based approaches; community leaderships; whole-service approaches; microfinancing; local adaptation; improvement of quality indicators and measures and accountability systems.
Tyler, I., et al., (2018). *It takes a village: a realist synthesis of social pediatrics program.*	Canada, USA, Europe, Australia	Peer-reviewed publication reporting on realist review	Children and families from structurally vulnerable, low-income communities	Child is viewed in context of society, neighborhood, and family. Four consistent patterns of care that may be effective in social pediatrics: (1) horizontal partnerships based on willingness to share status and power; (2) bridged trust initiated through previously established third party relationships; (3) knowledge support increasing providers’ confidence and skills for engaging community; and (4) increasing vulnerable families’ self-reliance through empowerment strategies.	Holistic focus; community participation and partnerships; intersectoral actions. Trauma-informed and strengths-based approaches, acknowledgement of family and community expertise; intersectoral collaboration and partnerships with providers, children, and families; sharing of power; relational approaches to care.
VicHealth. (2015). *Promoting equity in early childhood development for health equity through the life course*	Australia	Report synthesizing “current evidence”	Prenatal-8 years	Health and social policies that support health of parents, young children and the conditions in which families work and live; equitable access to healthcare and social care for families; interventions should be universal, but the level of support needs to be proportionate to need.	Collective approach to leadership and governance; community development; targeted neighborhood or geographic locations; universalistic with targeted interventions; intersectoral and cross sectoral actions; social participation and engagement and trust; universal primary care services alongside local-tailored and responsive service provision.
Wettergren, B., et al., (2016). *Child health systems in Sweden*	Sweden	Discussion paper in peer-reviewed publication	Prenatal to 18 years	Children and families involved in decision making, information systems and quality improvements. Integrated system of maternity, child, preschool and school health care that is mid-wife or nurse-led. National public health policies are supportive of parenting role, health promotion and universal outreach with extra support for structurally vulnerable families.	Comprehensive, integrated and responsive system integrating prenatal care with early years and early grades in school; focus on health promotion

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
