# Peer review of "Re-Envisioning an Early Years System of Care towards Equity in Canada: A Critical, Rapid Review"

_ijerph, 2022, doi:10.3390/ijerph19159594_

Round 1

Reviewer 1 Report

This is an interesting paper. However, there are points that need to be addressed before publication.

My main concerns are about methodology and how it was applied in the paper.

·         The authors claim that use the guide [58[ for their rapid review. But data extraction is not presented in detail. I propose the authors to create the data extraction table and include a sample of results from a review of studies

·  In the next step, the data synthesis, the authors have to use the results from the Data Extraction table to organize results and findings to simplify the process of drawing conclusions.

·         Additional tables can be created to group data and identify similarities and differences in results across studies and then discuss the topic as they do in the findings section.

Author Response

As we discuss in our methodology, rapid reviews are an emerging methodology (distinct from systematic reviews) in order to address in a timely way, the priorities of a stakeholder, which was the case in our review. In this re-submission we have included a data extraction section (highlighted in uploaded manuscript) and a data extraction table which provides a sample of papers that were reviewed as per the reviewer’s recommendation.

We have added the following:

It is beyond the scope of this paper to summarize all 130 manuscripts that were reviewed and a sample of papers from diverse global contexts are included in Table 2. This table highlights that despite the heterogenous and transdisciplinary nature of the publications reviewed, they each had clear recurring elements of an equity-orientation.

As noted, in this table, we have included a column to highlight what specific elements of an equity-orientation were evident in each of the sample publications. This is an extensive table and provides a clearer picture for your readers of how despite the heterogenous nature of the publications, there were clear and multiple elements of an equity-orientation in each paper. Many of these elements are recurring across publications and clearly align with and show up in the findings. This provides greater transparency between the data and our thematic analysis without the need for another separate table.

Reviewer 2 Report

Authors answered all my comments, no additional observations.

Author Response

Thank you and we have further improved our methodology as per your suggestion. 

In this re-submission we have included a data extraction section (highlighted in uploaded manuscript) and a data extraction table which provides a sample of papers that were reviewed as per the reviewer’s recommendation.

We have added the following:

It is beyond the scope of this paper to summarize all 130 manuscripts that were reviewed and a sample of papers from diverse global contexts are included in Table 2. This table highlights that despite the heterogenous and transdisciplinary nature of the publications reviewed, they each had clear recurring elements of an equity-orientation.

As noted, in this table, we have included a column to highlight what specific elements of an equity-orientation were evident in each of the sample publications. This is an extensive table and provides a clearer picture for your readers of how despite the heterogenous nature of the publications, there were clear and multiple elements of an equity-orientation in each paper. Many of these elements are recurring across publications and clearly align with and show up in the findings. This provides greater transparency between the data and our thematic analysis.

This manuscript is a resubmission of an earlier submission. The following is a list of the peer review reports and author responses from that submission.

Round 1

Reviewer 1 Report

The material is interesting, and the topic is relevant. However, it needs a considerable amount of work to be publishable. Some areas need clarification as noted below:

- Indicate the study’s design in the title and abstract.

- Overall organization and clarity throughout the manuscript should be improved. For the introduction, a restructuring of the writing to provide more coherent and connected ideas and sections would be valuable. Brief synopsis or syntheses of ideas and relationships between or within constructs would improve flow dramatically. Further, reorganized the manuscript sections in terms of the content they report. I suggest the following structure, namely: introduction (explain the general argument of the paper, without going into specific details) background (situate the study concepts within the context of extant knowledge, discuss the international relevance of the concepts) and purpose, creating greater clarity in the analysis of the reader.

- The article as it is constructed looks more like a reflection than a review. The main and fundamental purpose of writing a review is to create a readable synthesis of the best resources available in the literature for an important research question or a current area of research. Given the approach seems to be an integrative review, it is important to provide an explicit statement of questions being addressed with reference to population (or participants)/Concept/Context (PCC acronym).

- Describe the results of the search and selection process, from the number of records identified in the search to the number of studies included in the review, ideally using a flow PRISMA diagram.

- What is the size of the analysis corpus? It would be important to define whether this sample resulted in a basic research/ individual from each descriptor or is an advanced search and resulted therefore the intersection of different descriptors (e.g. Boolean method). Present full electronic search strategy, such that it could be repeated.

- Describe method of data extraction from reports (e.g., piloted forms, independently, in duplicate) and any processes for obtaining and confirming data from investigator.

- The description of the findings reads like thematic analysis. Therefore, the reader needs to understand how and why thematic methods can be used. This needs to be expanded, clarified, and supported by in-text citations.

- The results section is very dense and sometimes confusing, so it would benefit if a table were added with the key aspects.

- In the discussion section, there is a complete absence of the empirical implications of the study, besides which the theoretical implications should have been approached in greater depth;

- Please introduce study limitations. The limitations of the study could easily be addressed and incorporated within the discussion of this section (one important limitation is probably the heterogeneity of primary studies).

- Identify recommendations for practice/research/education/management as appropriate, and consistent with limitations, in order to more fully allow readers to understand the extent to which the authors were able to answer the research questions and to grasp the limitations of this study.

CHECKLIST FOR STYLE

Organization and style: The manuscript is clearly written and will serve a broad audience of students, researchers, and practitioners.

Reviewer 2 Report

This is an interesting paper that is worth to be published. However, there are some points to be addressed before publication.

My main concern about the paper is the methodology used. The paper is based on a systematic literature review but how the literature review was performed is not described in the paper. 

You can read for example:

Liberati, A., Altman, D. G., Tetzlaff, J., Mulrow, C., Gøtzsche, P. C., Ioannidis, J. P., ... & Moher, D. (2009). The PRISMA statement for reporting systematic reviews and meta-analyses of studies that evaluate health care interventions: explanation and elaboration. Journal of clinical epidemiology, 62(10), e1-e34.

Moher, D., Liberati, A., Tetzlaff, J., & Altman, D. G. (2010). Preferred reporting items for systematic reviews and meta-analyses: the PRISMA statement. Int J Surg, 8(5), 336-341.

Czeczotko, M., Górska-Warsewicz, H., & Zaremba, R. (2022). Health and Non-Health Determinants of Consumer Behavior toward Private Label Products—A Systematic Literature Review. International Journal of Environmental Research and Public Health, 19(3), 1768.

so to enhance the methodology section.

In the introduction, after describing the methodology give an overview of the entire paper to close the introduction.

I think that the paragraph "The authors contend that this critical ......... potentially preventable health inequities." should be removed from the introduction. 

Line 128 correct the year.

Reviewer 3 Report

Title: Orienting an Early Years System of Care towards Equity in a High-Income Country

Journal: International Journal of Environmental Research and Public Health

This paper has the potential to make a useful contribution to research on healthcare access.  There are however some important issues that need to be addressed. 

  1. Title: It is suggested to identify that this is a literature review.
  2. Introduction: The introduction focuses on a very specific group, the indigenous families. However, and although it may be the one group at the greatest disadvantage, there is a lack of data in terms of equity to support it. Additionally, it would be good to express how Canada stands in terms of health equity for children compared to other high-income countries or to those countries that have early year systems.
  3. Method: It is not clear when the study took place, it is mentioned in the paper June 202. The methodology needs more development, apart from specifying the dates when the study was carried out, how many articles were found? How was performed the article selection process, specify the inclusion and exclusion criteria? How were studies grouped for the synthesis? How many people participated reading or screening the articles? Any criteria or rationale to decide which articles were included in the proposed framework? How did you carry out the proposed framework? Any validation processes? etc.
  4. Results: results are general. Are there specific examples of effective early years systems that attends to child health equity issues that you consider are in some way feasible to implement in Canada? You lack a notion of cost versus benefit in this analysis, you may need to prioritize, any proposal in this sense?
  5. Conclusions: What are the limitations of this framework? Any limitations from the literature review? What are the main implications in terms of policy and future research?